# Solid-State Stability Profiling of Ramipril to Optimize Its Quality Efficiency and Safety

**DOI:** 10.3390/pharmaceutics13101600

**Published:** 2021-10-02

**Authors:** Katarzyna Regulska, Joanna Musiał, Beata J. Stanisz

**Affiliations:** 1Greater Poland Cancer Center, 15th Garbary Street, 61-866 Poznan, Poland; 2Department of Pharmaceutical Chemistry, Poznan University of Medical Sciences, Grunwaldzka Street, 60-780 Poznan, Poland; joanna.musial@student.ump.edu.pl

**Keywords:** ramipril, stability in solid state, excipients, tablets, degradation impurities

## Abstract

High global expenditure on out-of-label-date drugs, along with safety concerns associated with the accumulation of degradation impurities, justify the need for stability profiling. In this article, a comprehensive study on the solid-state stability of ramipril (RAM) was performed via isothermal methods under stress conditions. A validated stability-indicating HPLC protocol was used. The effects of various factors on the rate of RAM degradation were investigated, including: temperature, relative air humidity (RH), excipients (talc, starch, methylcellulose and hydroxypropyl methylcellulose), mode of tablet storage, and immediate packaging. The degradation impurities were also identified by HPLC–MS. It was found that RAM was unstable, and temperature accelerated its degradation. RAM was also vulnerable to RH changes, suggesting that it must be protected from moisture. The reaction followed first-order kinetics. The studied excipients stabilized RAM as a pure substance. The tableting process deteriorated its stability, explaining the need for appropriate immediate packaging. RAM in the form of tablets must be stored in blisters, and it cannot be crushed into two halves. The degradation impurities were ramiprilat and the diketopiperazine derivative.

## 1. Introduction

The stability profile of a pharmaceutical formulation is primarily defined by the physicochemical and microbial properties of the drug substance and drug product. Its assessment involves chemical identity and purity testing, physical appearance, and organoleptic testing, as well as microbiological limit testing. In particular, chemical stability is affected by the following factors: qualitative and quantitative composition of excipients and active ingredients, type of packaging, and environmental conditions, including temperature, relative air humidity (RH), and light. Comprehensive stability protocols must therefore evaluate the rate and mechanism of drug decomposition with respect to all of these variables so that drug shelf life under specified storage conditions can be accurately assessed. Stress tests seem the most appropriate research method in this regard; they provide an in-depth description of reaction kinetics and thermodynamics. Stress studies also allow the differentiation and qualification of all of the degradants—even those that might remain undetected under controlled real-life conditions. Essentially, the identification and quantitation of all degradation impurities is necessary not only for scientific but also for safety reasons. In fact, their gradual formation could translate into increased drug toxicity. Thus, the thorough understanding of all of the intrinsic processes that may lead to the deterioration of a pharmaceutical formulation during storage is indispensable for the development of a stable, effective, and safe final drug product [1,2].

Regulatory guidelines for the pharmaceutical industry provide extensive standard protocols for drug stability profiling (ICH Q1A–Q1F) and monitoring of impurities (ICH Q3A–Q3E, ICH M7). However, the relevant stability data remain confidential [3,4,5]. In general, commercially performed real-time stability assays set the shelf life of drugs conservatively in the range of 1–5 years. Typically, detailed stability studies are not performed by manufacturers, mainly due to their high cost and long duration. Furthermore, because of the financial interests of manufacturers, the shelf lives of drugs are officially claimed to be as short as possible. Underestimated expiry dates of pharmaceuticals become a financial challenge for healthcare systems. In fact, huge amounts of out-of-label-date drugs are disposed of annually, indicating that extending their expiration periods based on reliable scientific data would be a great benefit [1]. In fact, in 2019, Zilker et al. reported that the expiry period of a large number of pharmaceuticals can be prolonged even up to four times beyond their label expiry dates [1]. Furthermore, in 2009, it was reported that every USD 1 spent by the US Department of Defense on the extension of drug expiry dates resulted in savings of USD 94 dollars. This proves that safe prolongation of dug shelf life is possible and profitable [1,6]. In addition, despite the high level of pragmatism of the legally adopted International Council for Harmonisation stability testing guidelines, in 2018 they proved insufficient to control genotoxic impurities. This resulted in the N-nitrosodimethylamine (NMDA) crisis and a worldwide recall of sartans and ranitidine-containing products. At that time, N-nitrosoamine derivative—a potent carcinogenic impurity—was accidentally found in numerous drug formulations, probably as a degradation or in-process impurity; its concentrations significantly exceeded the acceptance criteria [7]. Since then, it has become clear that thorough and widely available knowledge on drug stability must be provided for both scientific and applicatory purposes. This claim seems the most relevant to pharmaceuticals used in long-term therapy, since patient exposure and the cumulative dose of their potentially toxic impurities may pose a significant safety concern.

Ramipril (RAM) is an angiotensin-converting enzyme inhibitor (ACE-I); it is widely used in the chronic treatment of many cardiovascular and renal-system-related diseases as first-line therapy; its beneficial, pleiotropic effects, manifested by a reduction in cardiac mortality, have been confirmed in numerous large-scale clinical trials (e.g., AIRE, HOPE, ONTARGET). Currently, it is considered to be one of the leading ACE-Is within the therapeutic class [8,9]. In 2018, the total number of RAM prescriptions in the US alone reached nearly 4 million [10], confirming its wide patient exposure. RAM is a prodrug intended for oral use, commercially formulated as tablets or capsules. Preliminary molecule screening suggests its high liability to hydrolytic degradation due to the presence of an ester bond. The probable clinical consequence of this reaction would be a reduction in its bioavailability. Based on the literature review, intramolecular cyclization as an alternative degradation pathway is also possible. This reaction was reported for other structurally-related ACE-Is, including imidapril (IMD) [11], moexipril (MOXL) [12], quinapril (QUIN) [13], enalapril (ENA) [14], lisinopril (LIS) [15], and perindopril (PER) [16]. However, the toxicological data on the potential products of RAM cyclization are still unknown. Similarly, the stability profile of RAM remains unavailable to date, and no validated stability-indicating analytical method has been described in the literature. Therefore, based on its long-term cardiovascular indications, RAM seems to be an ideal candidate for enhanced stability profiling—mainly for predictive and preventive purposes.

Taking all of the above issues into consideration, the aim of this multistage study was to provide detailed knowledge about the mechanism of chemical degradation of RAM under a variety of environmental conditions. The secondary objective was to establish the potential impact of the studied factors on the efficacy and safety of RAM use in real life. For that purpose, stability analysis and identification of degradation products were performed for pure RAM, its commercial tablets, and its model mixtures with common excipients. The corresponding reaction orders, degradation rate constants, and thermodynamic parameters were then determined using a validated stability-indicating HPLC method.

## 2. Materials and Methods

### 2.1. Materials

Pure RAM (100%) was purchased from Rolabo (Zaragoza, Spain, batch n°: 11.PT24.01.02). RAM in commercial tablets (dose 10 mg; batch n° H1606) was obtained from Sanofi-Aventis Deutschland GmbH, Frankfurt am Main, Germany. Talc (T), starch (S), methylcellulose (MC), and hydroxypropyl methylcellulose (HPMC) were purchased from Merck, Darmstadt, Germany. Methanol, acetonitrile, and potassium dihydrogen phosphate were purchased from Merck, Darmstadt, Germany.

### 2.2. Instruments

#### 2.2.1. HPLC Method

The stability study of RAM was performed by HPLC–UV using a LiChroCART^®^ column (250-4 HPLC-Cartridge, LiChrospher^®^ 100 RP-18, 250 mm × 4 mm × 5 μm, Merck, Darmstadt, Germany). The employed mobile phase consisted of acetonitrile/phosphate buffer (0.035 mol/L, pH = 2.0) at a ratio of 65:35 (*v*/*v*). The flow rate was 1.0 mL/min. The injection volume was 20 μL. The detection wavelength was set at 213 nm.

#### 2.2.2. HPLC-MS Method

The degradation products of RAM were identified by HPLC–MS using a LiChroCART^®^ column (250-4 HPLC-Cartridge, LiChrospher^®^ 100 RP-18, 250 mm × 4 mm × 5 μm, Merck, Darmstadt, Germany). The mobile phase consisted of methanol/water/formaldehyde (49:50:1, *v*/*v*/*v*). The flow rate was 0.5 mL/min. The injection volume was 100 μL. The range of registered masses was *m*/*z* 150 to 1000 or *m*/*z* 150 to 600. Positive (ES^+^) and negative (ES^−^) electrospray ionization modes were applied. A constant temperature of 298 K was maintained throughout the analysis.

#### 2.2.3. Other Instruments

The auxiliary equipment included a pH CD-401 electrode (Mettler Toledo, Greifensee Switzerland); a Sartorius BP 2105 analytical scale; and WAMED KBC—125 W thermal chambers with automatic temperature and relative humidity (RH) control (Wamed, Warsaw, Poland).

### 2.3. Procedures

#### 2.3.1. Preparation of the Mobile Phase

The mobile phase for HPLC–UV analysis was prepared by mixing acetonitrile and phosphate buffer (0.035 mol/L, pH = 2.0) at a ratio of 65:35 (*v*/*v*). Phosphate buffer was freshly prepared as follows: 0.0681 g of potassium dihydrogen phosphate (KH_2_PO_4_) was transferred to a 500 mL volumetric flask. Then, 400 mL of distilled water was added. The mixture was shaken until dissolved. Then, pH was adjusted to 2.0 using 85% phosphoric (V) acid. Finally, distilled water was added up to 500.0 mL.

To prepare the mobile phase for the HPLC–MS analysis, methanol, water, and formaldehyde were mixed at a ratio of 49:50:1 (*v*/*v*/*v*).

Both solutions were filtered using a filtration system equipped with a vacuum pump (filter 0.22 μm). Then, they were degassed using an ultrasonic cleaner for 20 min prior to use.

#### 2.3.2. Preparation of Pure RAM (Standard Solution)

First, 0.0100 g of pure RAM was weighed and transferred to a volumetric flask (volume of 25 mL). Then, 15 mL of methanol was added. The mixture was shaken until RAM was dissolved. Finally, it was filled with methanol up to 25.0 mL (solution B).

#### 2.3.3. Validation of HPLC for the Stability Assay of Pure RAM and RAM–Excipient Model Mixtures

##### Selectivity

First, 0.0100 g of pure RAM was weighed and transferred to a 5 mL glass vial. The sample was placed in a thermal chamber and heated (T = 363 K, RH = 76.0%). After 40 days, the sample was collected, cooled to room temperature, and its content was transferred to a volumetric flask (volume of 25 mL), using methanol as a solvent. The flask was shaken until the RAM dissolved. Then, it was filled with methanol up to 25.0 mL (solution A).

Next, 0.0100 g each of MC, HPMC, T and S were weighed to separate volumetric flasks (volume of 25 mL). Each flask was filled with 15 mL of methanol and shaken until its contents dissolved. The samples were diluted with methanol up to 25.0 mL and then filtered (solutions MC, HPMC, T, S).

Aliquots containing 20 μL of the solutions A, B (as described in Section 2.3.2), MC, HPMC, T and S were injected into the HPLC column. Chromatograms were registered.

##### Precision

Low and high precision levels were assessed. The exact amounts of 0.0050 g and 0.0100 g of RAM were weighed. The samples were transferred to eight volumetric flasks (volume of 25 mL), and methanol was added. The flasks were shaken for 15 min. Then, they were filled up with methanol to 25.0 mL (solutions P_Ni_ and P_Wi_). Solutions P_Ni_ and P_Wi_ were injected into the chromatographic column (20 μL). Chromatograms were registered. The obtained peak areas were subjected to statistical analysis.

##### Linearity

The calibration curve for RAM covered the concentration range from 0.004% to 0.040%. Exactly 0.0200 g of RAM was weighed and transferred to a volumetric flask (50 mL). The sample was dissolved in 50.0 mL of methanol (solution A). Aliquots of 1.0, 2.0, 3.0, 4.0, 5.0, 6.0, 7.0, 8.0, 9.0 and 10.0 mL of solution A were transferred to respective volumetric flasks. Then, they were diluted with methanol up to 10.0 mL (solutions A_i_).

The solutions A_i_ were injected into the chromatographic column (20 μL). Chromatograms were recorded. The obtained peak areas were analyzed as a function of RAM concentration.

##### Sensitivity

The limit of detection (DL) and the limit of quantification (QL) were calculated using the following formulae:(1)$$\mathrm{DL}=3.3\frac{S_{y}}{a}$$
(2)$$\mathrm{QL}=10\frac{S_{y}}{a}$$
where S_y_ is the standard deviation and a is the slope of the calibration curve [17].

#### 2.3.4. Kinetic Study—Preparation of Samples

An isothermal method was used to study the kinetic mechanism of solid-state degradation of RAM. The RH range was 50.0 to 76.0%, and the temperature range was 343 to 363 K. The experimental conditions were adjusted automatically. The experiments were performed for pure RAM, RAM in tablets and RAM in model mixtures, with the following excipients: T, S, HPMC and MC. The experimental conditions were achieved by introducing the samples into the thermal chambers that maintained the required temperature and RH levels throughout the experiments, with an accuracy of ± 1 K and ± 1%, respectively.

##### Pure RAM

A total of 0.0100 g of pure RAM was accurately weighed into 5 mL glass vials. For the evaluation of the thermal decomposition of RAM in solid state, the samples were exposed to the stress conditions of RH 76.0%, in a temperature range of 343—363 K. The maximum time of exposure was 150 days. To evaluate the effect of RH, the samples were exposed to stress conditions of T = 343 K and increasing RH of 50.0, 60.0, 66.0 and 76.0%. The maximum time of exposure was 100 days. The samples were collected and cooled to ambient temperature successively. The contents of the vials were quantitatively transferred to volumetric flasks using methanol. The flasks were shaken for 10 min; then, they were filled to 25.0 mL with the same solvent (solutions A_i_). The standard solution of RAM (solution B) was prepared as described in Section 2.3.2.

The solutions A_i_ and B were injected into the chromatographic column (20 µL). Chromatograms were registered. The content of pure RAM in the samples was calculated relative to the standard solution B.

##### RAM in Model Mixtures with Excipients

Model mixtures of RAM with the excipients MC, HPMC, S and T were prepared at a weight ratio of 1:1 (*w*/*w*). A total of 0.500 g of RAM and 0.500 g of the excipient were accurately weighed and transferred quantitatively to a porcelain mortar. The mixtures were homogenized for 15 min manually. Then, 0.0200 g of each model mixture was transferred to 5 mL glass vials.

To evaluate the thermal decomposition of RAM in model mixtures, the samples were exposed to stress conditions of RH 76.0% and increasing temperatures of 323 K, 328 K, 333 K and 343 K. The maximum time of exposure was 250 days. For the evaluation of the impact of RH, the samples of model mixtures were exposed to stress conditions of T = 343 K and increasing RH of 50.0, 60.0, 66.0 and 76.0%. The samples were collected and cooled to ambient temperature successively, at timepoints depending on the observed RAM decomposition rate. The maximum time of exposure was 200 days. The content of each vial was quantitatively transferred to 50 mL volumetric flasks, and then dissolved in 25.0 mL of methanol. The solutions were shaken for 30 min. Then, they were filtered through a quantitative 390 hard filter (Munktell) (solutions MC_i_, HPMC_i_. S_i_ and T_i_).

The standard solution of RAM (solution B) was prepared as described in Section 2.3.2.

Aliquots of 20 μL of solutions MC_i_, HPMC_i_, S_i_, T_i_ and B were injected into the chromatographic column. Chromatograms were registered. The concentration of RAM in the studied samples was calculated using the following formula:(3)$$C=\frac{P_{\mathrm{Ai}}\cdot c\cdot V\cdot M}{P_{B}\cdot m\cdot z}$$
where P_Ai_ (solutions MC, HPMC, S and T) and P_B_ represent the peak areas of the tested and the standard solutions of RAM, respectively; c is the concentration of RAM in the standard solution (%); V is the dilution factor; M is the model mixture mass; m is the weight of the sample and z is the content of RAM in the model mixture.

##### RAM in Tablets

Three isothermal experiments were performed: for whole blistered tablets, for whole blisterless tablets, and for halved blisterless tablets. In each experiment, samples with RAM tablets were placed in 5 mL glass vials and exposed to the stress conditions (temperature 318 K and RH 76.0%). After exposure, the contents of each vial were quantitatively transferred to 50 mL volumetric flasks. The test samples were disintegrated in 2.0 mL of distilled water and dissolved in 23.0 mL of methanol. The solutions were shaken for 15 min. Then, they were filtered through a quantitative 390 hard filter (Munktell) (solution A_i_). The standard solution of RAM (solution B) was prepared as described in Section 2.3.2.

Aliquots of 20 μL of solutions A_i_ and B were injected into the chromatographic column. Chromatograms were registered.

#### 2.3.5. Reaction Model Fitting

Mathematical model fitting was performed by comparing the correlation coefficients of theoretical curves. The following reaction models were considered: nucleation (power-law, Avrami–Erofeev, Prout–Tompkins), geometrical contraction (contracting area, contracting volume), diffusion (1D diffusion, 2D diffusion, 3D diffusion), and reaction-order (zero-order, first-order, second-order and third-order) [18].

#### 2.3.6. Calculation of Thermodynamic Parameters

For each experiment, the degradation rate constant (k) was determined. Then, its natural logarithm was plotted against the reciprocal of the corresponding temperature to fulfil the Arrhenius relationship:ln k_i_ = lnA − E_a_/RT(4)
where k_i_ is the reaction rate constant (s^−1^), A is the frequency coefficient, E_a_ is the activation energy (J mol^−1^), R is the universal gas constant (8.3144 J K^−1^ mol^−1^), and T is the temperature (K).

The activation energy (E_a_), enthalpy of activation (H^≠^), and entropy of activation (∆S^≠^) at 298 K and RH 76.4% were determined using the following equations:Ea = −a × R(5)
E_a_ = ΔH^≠^ + RT(6)
ΔS^≠^ = R lnA − ln KT/h(7)
where a is the slope of ln k_i_ = f(1/T), A is a frequency coefficient, E_a_ is the activation energy (J mol^−1^), R is the universal gas constant (8.3144 J K^−1^ mol^−1^), T is the temperature (K), ∆S^≠^ is the entropy of activation (J K^−1^ mol^−1^), ∆H^≠^ is the enthalpy of activation (J mol^−1^), K is the Boltzmann constant (1.3806488(13) × 10^−23^ J K^−1^), and h is Planck’s constant (6.62606957(29) × 10^–34^ J × s) [19].

#### 2.3.7. Identification of RAM Degradation Impurities by HPLC–MS

A total of 0.0100 g of pure RAM was weighed accurately into a 5 mL glass vial. The sample was stressed at temperature of 363 K and 76.0% RH for 40 h. The decomposed sample was collected and cooled to room temperature, and then it was quantitatively transferred to a volumetric flask using methanol. The flask was shaken until dissolved and filled up to 25.0 mL with methanol (solution A).

The standard solution of RAM (solution B) was prepared as described in Section 2.3.2.

Aliquots of 20 μL of solutions A and B were injected into the chromatographic column. Chromatograms were registered.

## 3. Results

### 3.1. Validation of the Analytical Method

The selectivity of the developed stability-indicating HPLC method was expressed by the retention times (t_R_) of the signals in the obtained chromatograms, as demonstrated in Figure 1. In the time range from 2 to 10 min, single and well-separated peaks were observed, which were attributed to RAM (t_R_ = 7.80 min) and RAM decomposition products (t_R_ = 4.15 and 5.83 min, respectively). The studied excipients (MC, HPMC, T, and S) exhibited no interference with RAM or its degradation impurities. Their peaks were not recorded in the tested time range (Figure 2). Standard deviation (SD), relative standard deviation (RSD) and variation coefficient (CV%) were used to evidence the method’s precision, as demonstrated in Table 1. Linearity was described by the parameters of the regression: correlation coefficient (r), slope (a) and intercept (b); they were determined by the least squares method. The regression parameters are provided in Table 2.

Under the applied analytical conditions, QL for RAM equaled 0.0083% and DL was 0.0027%.

### 3.2. Kinetic Parameters

#### 3.2.1. RAM in Pure Form and in Model Mixtures with S, T, HPMC and MC

Based on the obtained kinetic curves, the mechanism of solid-state RAM degradation in pure from and in model mixtures with S, T, HPMC, MC was assessed (Figure 3). The kinetic and thermodynamic parameters of RAM degradation are provided in Table 3 and Table 4. The effect of the studied excipient on the stability of RAM in the solid state under increasing RH is demonstrated in Figure 4, presented as a semi-logarithmic plot of reaction rate constants (k) versus RH%. The correlation lnk = f (RH%) was provided only for one model mixture, since in the remaining cases the curves were analogous.

#### 3.2.2. RAM in Tablets

The kinetic parameters of the decomposition of RAM in the final drug formulation were assessed for three different storage modes: tablets in commercial immediate packaging, whole tablets without commercial packaging, and bare tablets broken in half. The corresponding degradation rate constants (k) for pure RAM and for RAM in tablets under stress conditions (RH = 76.0%, T = 318 K) are depicted in Table 5.

#### 3.2.3. Identification of Degradation Impurities

The identification of pure RAM and its degradation impurities was performed by HPLC–MS. The obtained chromatograms and the mass spectra are shown in Figure 5. The remaining spectra are available in the Appendix A. The analysis of the obtained *m*/*z* values is demonstrated in Table 6. The identification of degradation impurities of RAM in tablets was also conducted via HPLC–MS. The obtained chromatogram is available in the Appendix A.

## 4. Discussion

The designed stability profiling was intended to investigate diverse potential sources of variability during RAM’s life cycle, covering the manufacturing process, logistics, storage, and real-life use. For this reason, the following factors that could affect its stability, efficiency, safety, and costs were considered: temperature, RH, excipients S, T, HPMC and MC, immediate packaging and real-life storage mode. The degradation impurities were also identified for safety reasons.

### 4.1. Validation Report

The initial stage of the performed studies was to develop and validate an appropriate stability-indicating method. This method had to be powerful enough to differentiate, qualify, and quantify RAM and its impurities. The procedure of choice was HPLC–UV, due to its wide accessibility, low cost, and the relative simplicity of its analytical process. The preliminary analytical conditions were selected based on a review of literature available for other structurally related ACE-Is [11,12,13,14,15,16,20,21,22,23,24]. Then, the protocol was optimized so as to achieve adequate levels of selectivity, precision, linearity, and sensitivity. The selectivity of the stability-indicating method is crucial for the monitoring of drug concentration changes in the presence of its degradants. This requirement is most significant, as these compounds usually share close physiochemical characteristics. The chromatograms obtained under the analytical conditions described in Section 2.3.3. Selectivity showed acceptable peak separation and sharp shape, meeting the selectivity criteria. The developed peaks were assigned to RAM and its two degradation products. There were no interferences from T, S, HPMC or MC, nor from the components of the tablet formulation (Figure 1 and Figure 2). The SD and RSD ranged from 0.0505 to 0.0146, and CV% was between 1.220% and 1.241% (Table 1), which was satisfactory. QL and DL were also accepted, suggesting good precision and sensitivity. Analysis of variance showed a linear relationship between peak areas and concentration. A calibration plot was determined and the corresponding regression equation was found (Figure 2, Table 2). A systematic error was excluded, indicating that the developed HPLC–UV protocol was applicable for the stability profiling of solid-state RAM in pure form; in model mixtures with S, T, HPMC, MC and in a tablet formulation.

### 4.2. Kinetic Order of RAM Degradation

The isothermal method under stress conditions of increased temperature and RH was applied for RAM stability profiling. It was selected due to its high exploratory potential to recognize all sources of variability that may affect the stability of RAM in solid state. This method is also an integral part of drug development and management programs in accordance with the quality by design concept, as per the ICH Q8 (R2) pharmaceutical development guideline. Determination of the kinetic order of RAM degradation was conducted by evaluating the percentage of the remaining drug content in the tested samples after their exposure to stress conditions at different time intervals. The stress conditions included increased temperature (343 to 363 K) and RH (50.0 to 76.0%). An analogous protocol had been adopted previously for other structurally related ACE-Is [12,13,14,15,16,22]. As expected, in the current study, a progressive loss of RAM over time was observed. Calculated concentrations in the successive samples were plotted versus time of exposure c_t_ = f(t) to construct kinetic curves. The reaction order was established mathematically via a model-fitting method. The highest correlation coefficient was obtained for the first-order reaction. Thus, it was discovered that solid-state degradation of RAM in pure form, as well as in the presence of S, T, MC and HPMC, is consistent with first-order kinetics. This was described by the following kinetic equation:(8)$${\mathrm{lnC}}_{t}=C_{0}-k\cdot t$$
where C_0_ and C_t_ represent the RAM concentration (%) at t_0_ and t_t_, respectively, k is the RAM decomposition rate constant in pure form and in model mixtures (s^−1^),and t is the time.

Based on the obtained linear plots lnC_t_ = f(t), the reaction rate constants (k) were determined; they corresponded to the slope (a) in the following manner: a = −k. The results confirmed that the degradation of RAM followed the most predictable kinetics, and that the studied excipients did not interfere with this process. In contrast, for other structurally related ACE-Is, the autocatalytic reaction model for their solid-state degradation was reported: IMD [11,20], MOEX [12], QUIN [13], ENA [14], LIS [15] and PER [16]. The half-life of pure RAM was calculated using the following formula: t_0,5_ = 0.693/k; under the conditions of RH 76% and T = 363 K it equaled 146 h. Considering the half-life values within the therapeutic class, ACE-Is can be arranged in the following order: QUIN > PER > MOEX > ENA > cilazapril (CIL) > RAM > IMD > LIS > benazepril (BEN). BEN is the most stable as its t_0,5_ is the longest (266 h) [11,12,13,14,15,16,20,21,22,23,24,25].

### 4.3. Effect of Temperature

The effect of heating on the degradation rate of RAM in pure form and in model mixtures was examined. The experiments were performed at four temperatures: 343 K, 348 K, 353 K, and 363 K. Kinetic plots were constructed for each series of samples, and the degradation rate constants (k) were elucidated. The obtained results were plotted so as to fulfil the Arrhenius relationship lnk_i_ = f(1/T) (Table 3). This clearly demonstrated that stress conditions of increased temperature significantly compromised RAM’s stability in all experiments, as expressed by the increase in degradation rate constants (k) with temperature. It was also confirmed that heating did not change the kinetic mechanism of RAM degradation.

According to the transition state theory, the thermodynamic parameters of RAM degradation in pure form and in the presence of excipients were estimated. The value of E_a_ describes the minimum energy input necessary to initiate the reaction; this corresponds to the strength of the bonds split. The lowest E_a_ was obtained for pure RAM (E_a_ = 96.39 ± 22.4 (KJ/mol)), indicating its greatest vulnerability to degradation in this form. The studied excipients stabilized RAM in the following order: T > HPMC > MC > S, as evidenced by increasing E_a_ in the respective model mixtures (Table 3). This demonstrates that the tested excipients are suitable for commercial RAM formulations. The positive ΔH^≠^ in each experiment indicated that the observed reactions of RAM degradation were endothermic—they needed a constant energy supply for the formation of the activated complex. The ∆S^≠^ were negative, which is characteristic of bimolecular reactions.

Summarizing the E_a_ values within a therapeutic class, ACE-Is can be ranked in the following order: RAM > IMD > MOEX > BEN > PER > QUIN > ENA > LIS > CIL, indicating that RAM is the most vulnerable to temperature changes [11,12,13,14,15,16,20,21,22,23,24,25].

### 4.4. Effect of RH

The effect of RH on the degradation rate constant (k) of RAM in pure form and in the presence of T, S, MC and HPMC was investigated at T = 343 K, in an RH range of 25.0–76.0%. Linear relationships lnk_i_ = f (RH%) were developed for each experiment. This confirmed that humidity did not change the kinetic mechanism of RAM degradation in pure form or in the presence of excipients; however, it increased its rate exponentially. The magnitude of slope for each relationship lnk_i_ = f (RH%) defines the vulnerability of RAM to RH-dependent degradation (Figure 4). Pure RAM was found to be the most sensitive, while the excipients stabilized it in the following order: T > MC > S > HPMC. Based on the literature, the sensitivity to RH variations within the ACE-I class increases as follows: BEN > ENA > IMD > CIL > QCHL = RAM > MOXL [11,12,13,14,15,16,20,21,22,23,24,25].

### 4.5. Three-Dimensional Correlation between T, RH, and k for RAM Degradation under Humid Conditions

The achieved experimental data on the solid-state stability of pure RAM versus RH and temperature were demonstrated collectively as a three-dimensional stability surface, which was described by the following relationship:lnk = −5610.72 (1/T) + 0.067 (RH) − 4.00(9)

The proposed 3D correlation was based on linear semi-logarithmic plots: lnk_i_ = f (RH%) and lnk_i_ = f (1/T). The surface is convenient and practical for predictive purposes (Figure 6); it can facilitate the calculation of the degradation rate constant for solid-state RAM decay by the use of easy-to-measure parameters of drug storage.

### 4.6. RAM in Tablets

Experiments with RAM in final dosage forms were performed for three tablet series: whole blistered tablets, whole blisterless tablets and halved blisterless tablets. Their aim was to investigate the effects of drug formulation processes and in-home storage habits on drug quality and safety.

In our studies, it was confirmed that the process of tableting did not influence the mechanism of RAM decomposition, since first-order kinetics were observed:(10)$${\mathrm{lnC}}_{t}=C_{0}-k\cdot t$$
where C_0_ and C_t_ represent the RAM concentration (%) at t_0_ and t_t_, respectively, k is the RAM decomposition rate constant in commercial tablets (s^−1^) and t is the time.

However, tableting caused an acceleration of RAM degradation when compared to the pure substance, as evidenced by a higher degradation rate constant for RAM in the blisterless tablet series (Table 6). Regarding storage habits, it was found that halved tablets without blisters exhibited the poorest stability profile. Similarly, the stability of whole blisterless tablets was compromised. The most stable were whole blistered tablets, evidencing that this is the only proper mode of drug handling. This information must be clearly explained to patients by the dispensing pharmacists.

### 4.7. Degradants

The identification of RAM degradation products via HPLC–MS facilitated the description of its decay pathways. The obtained chromatogram for pure RAM depicted one peak at t_R_ ≈ 6.85 min for the anionic form ES^−^ and t_R_ ≈ 6.98 for the cationic form ES^+^; the *m*/*z* values were 415 and 417, respectively. This confirmed the identity of the drug (RAM molar mass = 416). The HPLC–MS chromatogram registered for the degraded sample (after stress at T = 363 K, RH = 76.4% for 40 days) revealed three peaks attributed to the parent compound and two degradation impurities: (a) RAM (t_R_ ≈ 6.88 min for ES^−^ and t_R_ ≈ 6.86 for ES^+^; *m*/*z* = 415 and 417, respectively); (b) degradation impurity n°1 (t_R_ ≈ 3.38 min for ES^−^ and t_R_ ≈ 3.45 min for ES^+^; *m*/*z* = 387 and 389, respectively), and (c) degradation impurity n°2 (t_R_ ≈ 5.41 min for ES^−^ and t_R_ ≈ 5.49 min for ES^+^; *m*/*z* = 397 and 399, respectively); this is demonstrated in Figure 5. Based on the obtained m/z values, it was concluded that pure RAM underwent simultaneous hydrolysis with the formation of ramiprilat (RAMat), and intramolecular cyclization that produced the diketopiperazine derivative (DKP) (Table 6). Under the applied experimental conditions, the dominant reaction was hydrolysis over cyclization, as evidenced by the 20-fold larger HPLC and HPLC–MS peak sizes for RAMat than for DKP (Figure 5 and Figure 7). RAMat is the active form of RAM; it has low toxicity; however, it is not absorbed from the gastrointestinal tract following oral administration; thus, its formation compromises RAM’s bioavailability and potency. The presence of DKP, in turn, may pose a safety concern, as the data on its toxicity remain unavailable until now. Therefore, owing to RAM’s indications for long-term use and its potential high lifetime exposure, further toxicological studies are recommended for DKP.

The HPLC–MS chromatogram obtained for degraded RAM in tablets demonstrates one distributed peak, which was assigned to RAMat (t_R_ ~ 3.49 min) (Appendix A). This suggests that under the applied conditions, the degradation of the drug in tablets was mainly by de-esterification.

## 5. Conclusions

Based on the present research, it can be concluded that RAM is liable to thermal degradation, and that the main factor accelerating this process is RH. Therefore, RAM as a substance must be stored in closed containers protected from moisture, while tablets should be stored in immediate packaging. With this respect, patients must be clearly informed by label claims and dispensing pharmacists that storing halved RAM tablets or tablets removed from the blister (e.g., in pill organizers) shortens their shelf life significantly, and this mode of storage should generally be avoided. The tableting process deteriorates RAM’s stability compared to the pure substance. Therefore, appropriate immediate packaging for final drug formulation is obligatory. The stabilization of RAM can be achieved by co-formulation with S, T, HPMC and MC, since their presence is considered to protect the ester bond in the RAM molecule from hydrolysis. Thus, these excipients are recommended for RAM manufacture as first-choice excipients. The degradation of RAM follows parallel pathways of hydrolysis and intramolecular cyclization, with the former predominating. Hydrolysis compromises the bioavailability of RAM, while the clinical effect of intramolecular cyclization remains unknown, and requires further study.

Considering all of the issues discussed, the novelty of this work can be summarized as a comprehensive elucidation of the kinetic mechanism of RAM’s thermal degradation for the first time. These data provide a reliable scientific basis for extending the shelf life or confirming the quality of RAM tablets and pure RAM after exposure to various environmental conditions, and potentially in real-life storage. Thus, the minimization of drug waste can be achieved, which would be an effective cost reduction strategy. Furthermore, since the effects of various excipients on the stability of an active ingredient are not homogenous in general, the data presented herein clearly indicate which excipients should be considered by manufacturers as RAM stabilizers during the formulation development. Moreover, for the first time it was confirmed that no toxic impurities are formed during RAM’s storage. This knowledge became an important safety issue after the global N-nitrosamine crisis in 2018, as the regulations in force at that time did not address this subject in sufficient detail. Finally, for the dispensing pharmacists, clear confirmation of the best in-home storage practice provided here is useful for increasing patient compliance with label claims. This aspect seems crucial, as self-verification of excessive RAM loss during improper in-home storage is impossible due to the absence of any physical changes in the tablets’ appearance.

## Figures and Tables

**Figure 1 pharmaceutics-13-01600-f001:**
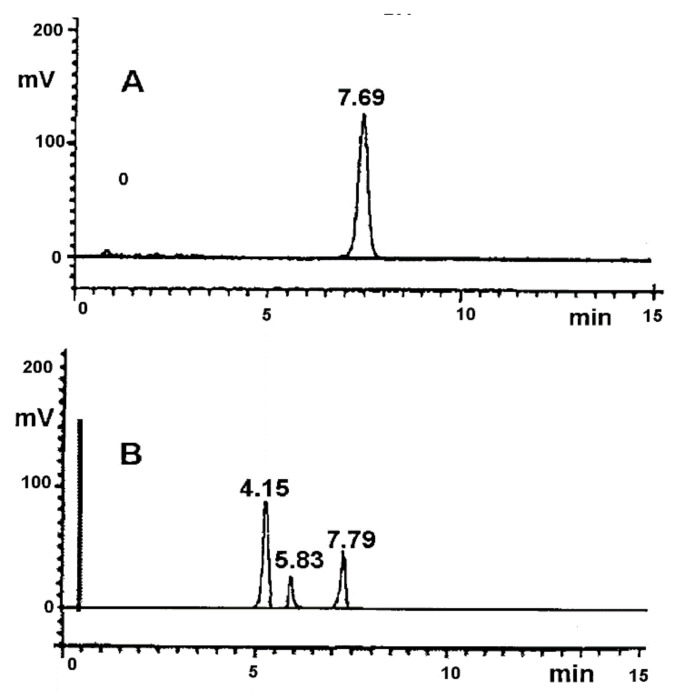
HPLC chromatograms: (**A**) pure RAM in non-degraded sample; (**B**) pure RAM after exposure to stress conditions (T = 363 K, RH = 76.0%, t = 40 days).

**Figure 2 pharmaceutics-13-01600-f002:**
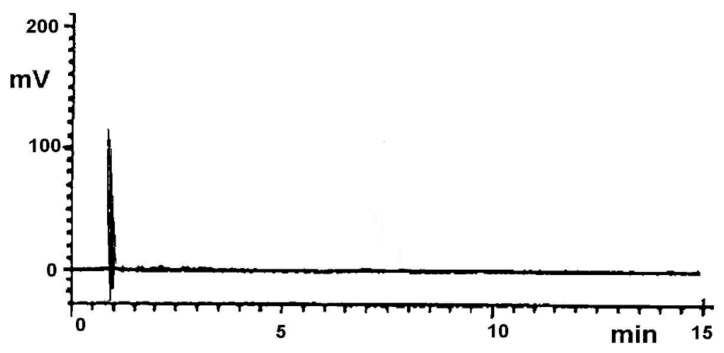
HPLC chromatogram of RAM–excipient model mixture solution.

**Figure 3 pharmaceutics-13-01600-f003:**
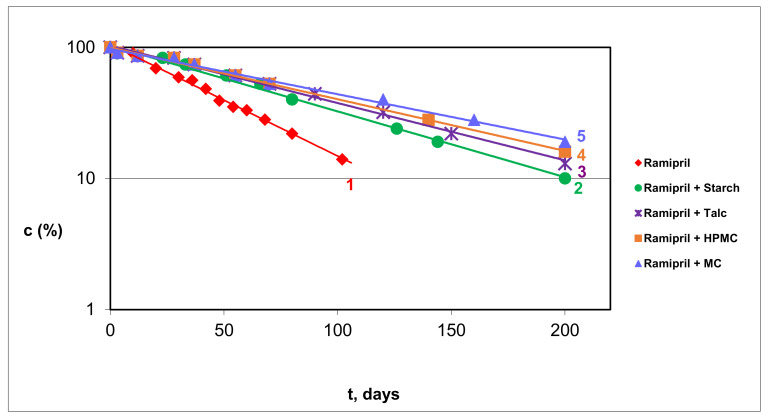
Kinetic curves for RAM decomposition at T = 343 K and RH = 76.0%: (1) pure RAM, (2) RAM–S, (3) RAM–T, (4) RAM–HPMC, (5) RAM–MC. Changes in RAM concentration as a function of time.

**Figure 4 pharmaceutics-13-01600-f004:**
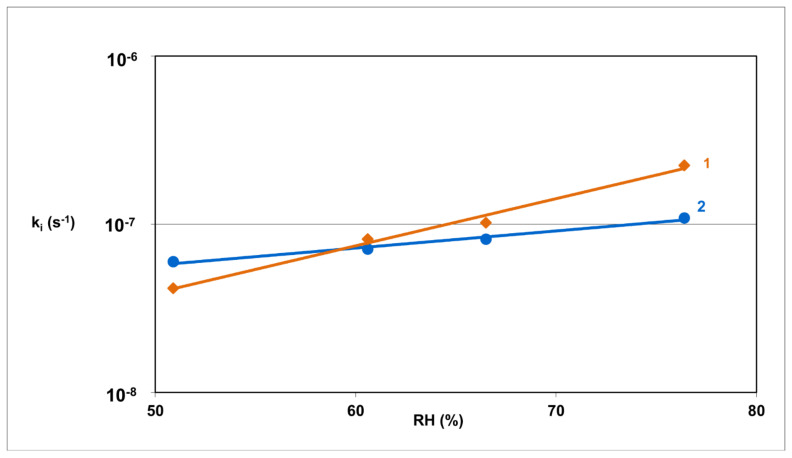
Semi-logarithmic plots of the degradation rate constant of pure RAM and RAM in the presence of HPMC as a function of RH% under T = 343 K (1: pure RAM, 2: RAM–HPMC).

**Figure 5 pharmaceutics-13-01600-f005:**
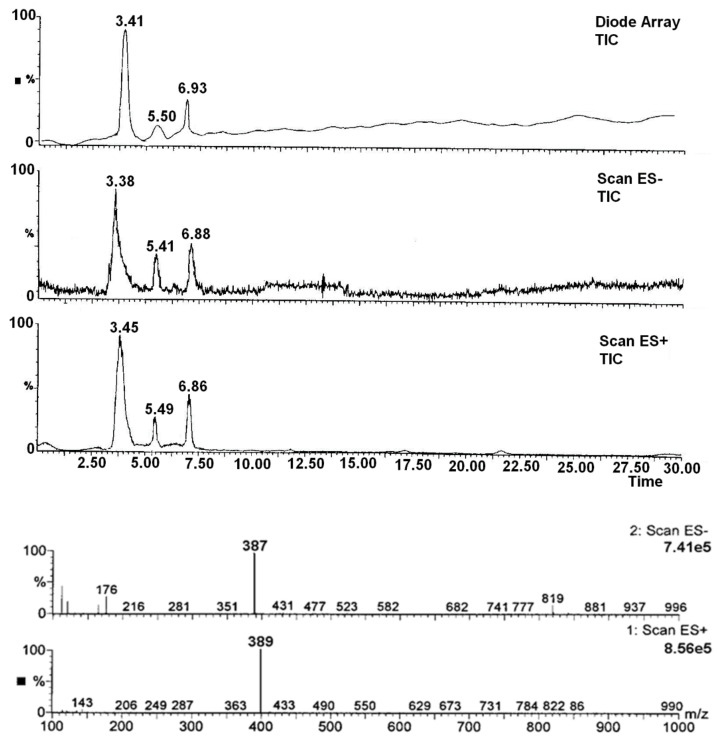
HPLC–MS chromatograms and mass spectra for solid-state RAM in degraded sample (T = 363 K, RH = 76%): identification of degradants.

**Figure 6 pharmaceutics-13-01600-f006:**
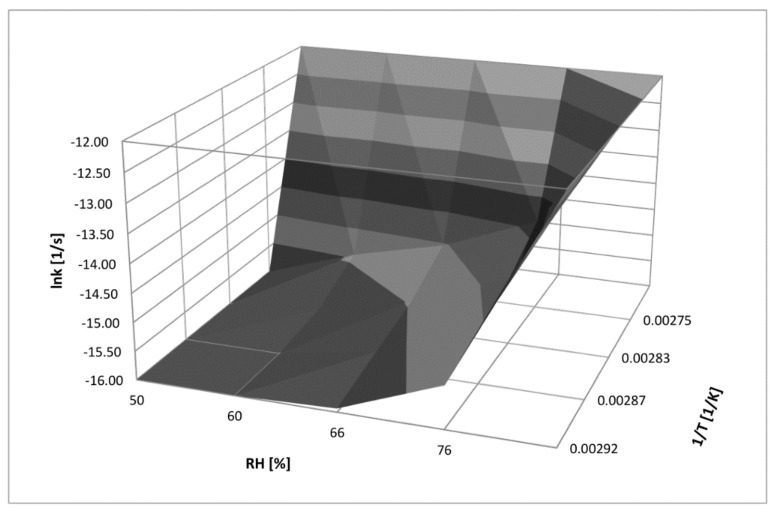
Three-dimensional relationship between temperature (1/T), RH, and k (lnk) for solid-state RAM degradation under humid conditions.

**Figure 7 pharmaceutics-13-01600-f007:**
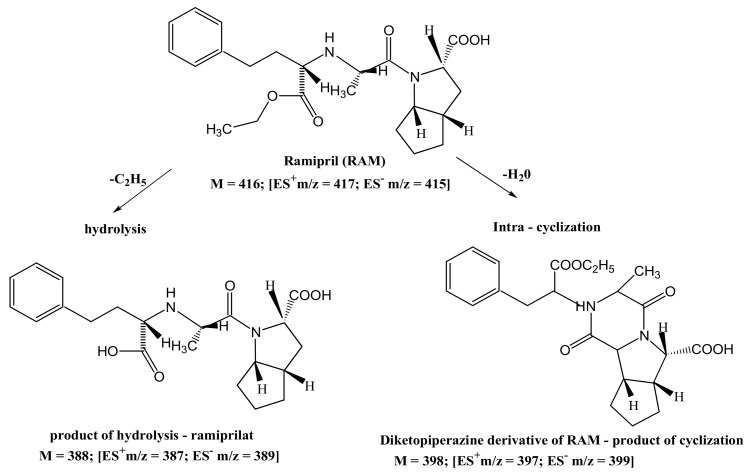
Proposed degradation pathway of RAM in the solid state.

**Table 1 pharmaceutics-13-01600-t001:** Precision of the stability-indicating HPLC method for RAM analysis: low and high levels.

Statistical Evaluation	Low Precision Level	High Precision Level
Arithmetic mean of the measured value, area	3.4489	6.8340
Standard deviation (SD)	0.0505	0.0792
Relative standard deviation (RSD)	0.0146	0.0124
Variation coefficient (CV%)	1.2200	1.2410

**Table 2 pharmaceutics-13-01600-t002:** Regression parameters for the stability-indicating HPLC method for RAM analysis.

Statistical Analysis for y = ax + b	Statistical Analysis for y = ax
a ± Δa = 138.87 ± 7.62	a ± Δa = 144.22 ± 7.05
b ± Δb = 0.149 ± 0.189	S_a_ = 3.119
S_a_ = 3.308	S_y_ = 0.113
S_b_ = 0.0821	r = 0.998
S_y_ = 0.120	
r = 0.998	

The relevance of the coefficient b in P_i_/P_Wz_ = f(c) shows that b = 0, as t_b_ = b/S_b_ = 1.8246, which is lower than the critical value t_0.05_ = 2.3060.

**Table 3 pharmaceutics-13-01600-t003:** Effect of temperature on the stability of pure RAM and RAM in model mixtures with excipients (RH 76.0%): kinetic and thermodynamic parameters.

T (K)	Decomposition Rate Constants in Solid Phase k_i_ (s^−1^)	Parameters of the Arrhenius Relationship k_i_ = f(1/T)	Thermodynamic Parameters of RAM Decomposition (RH 76%)
RAM in pure			
343	(2.241 ± 0.185) × 10^−7^	a = −5138.0 ± 561.2	E_a_ = (96.39 ± 22.4) KJ/mol
348	(3.035 ± 0.293) × 10^−7^	b = 1.032 ± 0.417	∆H^≠^ = (93.91 ± 24.8) KJ/mol
353	(5.145 ± 0.351) × 10^−7^	S_a_ = 176.360	∆S^≠^ = (−92.01 ± 189.2) KJ/(K mol)
363	(1.343 ± 0.093) × 10^−6^	S_b_ = 0.449	T = 298 K
		S_y_ = 0.449	
		r = −0.998	
RAM–MC model mixture			
323	(5.860 ± 0.122) × 10^−9^	a = −13580.2 ± 4041.5	E_a_ = (112.91± 33.6) KJ/mol
328	(9.900 ± 0.149) × 10^−9^	b = 23.74 ± 10.72	∆H^≠^ = (110.43 ± 36.7) KJ/mol
333	(5.141 ± 0.168) × 10^−8^	S_a_ = 1270.110	∆S^≠^ = (−47.57 ± 98.5) KJ/(K mol)
343	1.151 ± 0.149) × 10^−7^	S_b_ = 3.862	T = 298 K
		S_y_ = 0.223	
		r = −0.991	
RAM–HPMC model mixture			
323	(4.890 ± 0.480) × 10^−9^	a = −13976.4 ± 3528.2	E_a_ = (116.20 ± 29.3) KJ/mol
328	(9.168 ± 0.254) × 10^−9^	b = 24.83 ± 9.36	∆H^≠^ = (113.71 ± 31.8) KJ/mol
333	(4.569 ± 0.388) × 10^−8^	S_a_ = 1108.800	∆S^≠^ = (−38.47 ± 167.2) KJ/(K mol)
343	(1.090 ± 0.286) × 10^−7^	S_b_ = 3.371	T = 298 K
		S_y_ = 0.195	
		r = −0.993	
RAM–S model mixture			
323	(4.660 ± 0.268) × 10^−9^	a = −14824.4 ± 2433.1	E_a_ = (123.25 ± 20.2) KJ/mol
328	(8.161 ± 0.396) × 10^−9^	b = 27.38 ± 6.45	∆H^≠^ = (120.78 ± 22.7) KJ/mol
333	(4.154 ± 0.816) × 10^−8^	S_a_ = 764.64	∆S^≠^ = (−17.26 ± 109.2) KJ/(K mol)
343	(1.268 ± 0.452) × 10^−7^	S_b_ = 2.325	T = 298 K
		S_y_ = 0.134	
		r = −0.997	
RAM–T model mixture			
323	(8.264 ± 0.268) × 10^−9^	a = −11910.2 ± 2248.9	E_a_ = (99.02 ± 18.7) KJ/mol
328	(1.264 ± 0.665) × 10^−8^	b = 18.42 ± 5.96	∆H^≠^ = (101.51 ± 21.2) KJ/mol
333	(4.268 ± 0.426) × 10^−8^	S_a_ = 706.760	∆S^≠^ = (−89.43 ± 105.2) KJ/(K mol)
343	(1.152 ± 0.338) × 10^−7^	S_b_ = 2.149	T = 298 K
		S_y_ = 0.124	
		r = −0.996	

**Table 4 pharmaceutics-13-01600-t004:** The influence of RH on the stability of RAM in pure form and in the presence of the excipients MC, HPMC, S, and T; T = 343 K.

RH (%)	Decomposition Rate Constants (k ± ∆k) s^−1^ for RAM in the Presence of Excipients			
	MC	HPMC	S	T
50	(7.158 ± 0.461) 10^−8^	(5.998 ± 0.328) 10^−8^	(7.234 ± 0.234) 10^−8^	(8.518 ± 0.398) 10^−8^
60	(8.145 ± 0.435) 10^−8^	(7.129 ± 0.569) 10^−8^	(8.657 ± 0.781) 10^−8^	(9.551 ± 0.670) 10^−8^
66	(9.216 ± 0.541) 10^−8^	(8.161 ± 0.871) 10^−8^	(9.551 ± 0.443) 10^−8^	(1.005 ± 0.981) 10^−7^
76	(1.157 ± 0.781) 10^−7^	(1.097 ± 0.563) 10^−7^	(1.268 ± 0.542) 10^−7^	(1.149 ± 0.894) 10^−7^
	Regression Parameters ln k_i_ = a (RH%) + b			
a ± ∆a	0.0187 ± 0.0053	0.0234 ± 0.0068	0.0215 ± 0.0068	0.0116 ± 0.0022
b ± ∆b	−17.43 ± 0.35	−17.86 ± 0.43	−17.56 ± 0.44	−16.89 ± 0.14
S_a_	0.0017	0.0021	0.0022	0.0007
S_b_	0.108	0.137	0.136	0.045
r	0.992	0.991	0.990	0.996
**RH (%)**	**Decomposition rate constants (k ± ∆k) s^−1^ for pure RAM**		**Regression parameters** **ln k_i_ = a(RH%) + b**	
50	(4.166 ± 0.885) 10^−8^		A ± ∆a	0.0646 ± 0.0152
60	(8.169 ± 0.982) 10^−8^		B ± ∆b	−20.29 ± 0.98
66	(1.022 ± 0.465) 10^−7^		S_a_	0.0048
76	(2.248 ± 0.664) 10^−7^		S_b_	0.307
			r	0.995

**Table 5 pharmaceutics-13-01600-t005:** Kinetic parameters of solid-state RAM decomposition in final drug formulation.

	(k ± ∆k) s^−1^	t_0.1_ (days)	t_0.5_ (days)
RAM in pure form	(1.449 ± 0.156) × 10^−8^	84.2	553.5
RAM tablets in blisters	(0.993 ± 0.846) × 10^−8^	123.2	810.2
RAM in whole blisterless tablets	(1.865 ± 0.795) × 10^−8^	65.2	428.8
RAM in halved blisterless tablets	(4.395 ± 0.328) × 10^−8^	27.8	182.7

k: reaction rate constants; t_0.5_: half-life; t_0.1_: shelf life.

**Table 6 pharmaceutics-13-01600-t006:** Identification of RAM decomposition products (T = 363 K, RH = 76%) by the analysis of *m*/*z* values from mass spectra.

Compound	Molecular Formula	Molecular Mass	*m*/*z* in ES^+^ [M-H]^+^ Ion	*m*/*z* in ES^−^ [M-H]^−^ Ion
RAM	C_23_H_32_N_2_O_5_	416	417	415
Product n°1 (RAMat *)	C_21_H_28_N_2_O_5_	388	387	389
Product n°2 (DKP-RAM) **	C_23_H_30_N_2_O_4_	398	397	399

* RAMat = ramiprilat; ** DKP-RAM = diketopiperazine derivative of RAM.

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
