# Peer review of "Solid-State Stability Profiling of Ramipril to Optimize Its Quality Efficiency and Safety"

_pharmaceutics, 2021, doi:10.3390/pharmaceutics13101600_

Round 1
Reviewer 1 Report
In my opinion, this manuscript does not fit the scope of the journal. It is really focused on analytical aspects and the degradation profile of ramipril. Minimal content regarding the pharmaceutical perspective is included. Also, the English needs revision. Sentences should not start with a number, many sentences need to be broken down to be easily understood and so on.
In addition, some parts of the result section should be moved on into methods (such as the equations utilised for LOD and LOQ in line 270 and so on).
Author Response
Point 1. In my opinion, this manuscript does not fit the scope of the journal. It is really focused on analytical aspects and the degradation profile of ramipril. Minimal content regarding the pharmaceutical perspective is included. Also, the English needs revision. Sentences should not start with a number, many sentences need to be broken down to be easily understood and so on. 

Response 1: Dear Reviewer we believe that the subject of our study fits the scope of the journal as several similar publications are already approved by the Editors and published online:
Alexander Benet et al. The Effects of pH and Excipients on Exenatide Stability in Solution. Pharmaceutics 2021, 13(8), 1263; https://doi.org/10.3390/pharmaceutics13081263
Sonal V. Bhujbal et al. Effect of Storage Humidity on Physical Stability of Spray-Dried Naproxen Amorphous Solid Dispersions with Polyvinylpyrrolidone: Two Fluid Nozzle vs. Three Fluid Nozzle. Pharmaceutics 2021, 13(7), 1074; https://doi.org/10.3390/pharmaceutics13071074
Nika Osel et al. Stability-Indicating Analytical Approach for Stability Evaluation of Lactoferrin. Pharmaceutics 2021, 13(7), 1065; https://doi.org/10.3390/pharmaceutics13071065
Jumpei Saito et al. Stability of Hydrocortisone in Oral Powder Form Compounded for Pediatric Patients in Japan. Pharmaceutics 2021, 13(8), 1267; https://doi.org/10.3390/pharmaceutics13081267
Žane Temova Rakuša et al. Comprehensive Stability Study of Vitamin D3 in Aqueous
Solutions and Liquid Commercial Products. Pharmaceutics 2021, 13(5), 617; https://doi.org/10.3390/pharmaceutics13050617 - 25 Apr 2021
Ema Kosović et al. Stability Study of Cannabidiol in the Form of Solid Powder and Sunflower Oil Solution. Pharma-ceutics 2021, 13(3), 412; https://doi.org/10.3390/pharmaceutics13030412
Point 2. English needs revision. Sentences should not start with a number, many sentences need to be broken down to be easily understood and so on.
Response 2. All text has been revised, English corrected.
Point 3. In addition, some parts of the result section should be moved on into methods (such as the equations utilised for LOD and LOQ in line 270 and so on).
Response 3. Corrected.
Reviewer 2 Report
I have reviewed through the whole manuscript critically and I found that authors have attempted a very good work on “solid state stability study on ramipril for efficiency, safety and cost”. Below please find my comments:
Line 265, 268 and line 475: Error! Reference source not found, please double check.
Line 281, for number decimals in table 1, please be consistent.
Line 430, typo, most liable “to” RH changes
Line 446, for figure 8, lnk [1/s] is overlapped with numbers, please modify the 3D figure.
Line 462, typo, the “latter” one is the only…
Author Response
Dear Reviewer! Thank you for your comments. We revised our manuscript accordingly.
Point 1. Line 265, 268 and line 475: Error! Reference source not found, please double check.
Response 1. Corrected.
Point 2. Line 281, for number decimals in table 1, please be consistent.
Response 2. Corrected.
Point 3. Line 430, typo, most liable “to” RH changes.
Response 3. Corrected.
Point 4. Line 430, typo, most liable “to” RH changes
Response 4. Corrected.
Point 5. Line 446, for figure 8, lnk [1/s] is overlapped with numbers, please modify the 3D figure.
Response 5. Corrected.
Point 6. Line 462, typo, the “latter” one is the only…
Response 6. Corrected.

Reviewer 3 Report
The manuscript by Regulska et al, investigated the impact of storage conditions, excipients, mode of tablet storage and packing on the stability of the ramipril degradation. In general, the study was well designed and written. The chemical degradants of ramipril at various storage conditions were determined using a validated HPLC method. However, no studies were conducted to determine the interaction between the drug and excipients tested and the solid-state nature of the drug. I would recommend authors to address this comment and following minor comments before accepting the manuscript.
Line 16: In scientific writings, the names with registered symbol or trademark written in capitalize each word font case. Correct the font case of ‘Ramipril’ wherever it has mentioned.
Line 73: Is it ‘label data’ or ‘label date’?
Line 118: Mention the purity of ramipril.
Section 2.1: Add the materials source location.
Section 2.3.4.2: Mention the duration of exposure of samples to the mentioned stability conditions.
Figures: Centre the positions of axis labels in the figures.
Figure 8: Axis labels overlapped with the numbers. Correct.
Author Response
Dear Reviewer! Thank you for your comments. We revised our manuscript accordingly.
Point 1. However, no studies were conducted to determine the interaction between the drug and excipients tested and the solid-state nature of the drug. I would recommend authors to address this comment and following minor comments before accepting the manuscript.
Response 1. Dear Reviewer, thank you for this good point. In fact we performed the DSC analysis for pure drug and drug-excipient model mixtures and the obtained thermograms demonstrated no physical interaction. We had a very thorough discussion about including these results into this publication. However, since we intended to focus our manuscript on chemical not physcial stability of ramipril in different forms we considered that these data would be beyond its scope.
Point 2. Line 16: In scientific writings, the names with registered symbol or trademark written in capitalize each word font case. Correct the font case of ‘Ramipril’ wherever it has mentioned.
Response 2. Corrected.
Point 3. Line 73: Is it ‘label data’ or ‘label date’?
Response 3. Corrected to ‘label expiry date’.
Point 4. Line 118: Mention the purity of ramipril.
Response 4. Corrected (100%).
Point 5. Section 2.1: Add the materials source location.
Response 5. Corrected – Poland.
Point 6. Section 2.3.4.2: Mention the duration of exposure of samples to the mentioned stability conditions.
Response 6. In section 2.3.4.2 the maximum time of exposure was provided.
Point 7. Figures: Centre the positions of axis labels in the figures.
Response 7. Corrected.
Point 8. Figure 8: Axis labels overlapped with the numbers. Correct.
Response 8. Corrected.

Reviewer 4 Report
This is a stability study, conducted at different relative humidity levels and temperatures. There is no direct connection of this work and the costs of cardovascular pharmacotherapy, therefore the title is not justified by the content of the manuscript, and the first paragraph of the introduction is rather irrelevant (unless the authors provide some data or some model based estimate of the impact of ramipril's degradation on the costs of pharmacotherapy).
There are also some other points that the authors need to consider before the manuscript can be accepted for publication, as listed below:
- The materials and methods does not contain a clear description of the methods, but rather a reference to the instrumentation.
- A description of how the relative humidity was regulated is missing.
- Also, a description of the model fitting procedure should be added. How many and which kinetic models did the authors fit and why did they choose the first order model as the best fitting one?
- A description of the transition state theory equations should be added in the methods, not in the results.
- Confidence intervals are necessary in figure 4, as the lines overlap significantly.
- Are Figures 5, 6 and 7 necessary? They can be moved to the supplementary information.
- As already mentioned, a better description of the transition state theory is needed. It is not clear how the activation energy is derived, by fitting the Arrhenious equation or by approximations of the parameters of transition state theory equations.
- What exactly is the novelty of this work? It is not unexpected that heat and moisture promote degradation of an API, neither it is unexpected that in presence of excipients, APIs degrade faster.
Author Response
Dear Reviewer! Thank you for your comments. We revised our manuscript accordingly.
Point 1. This is a stability study, conducted at different relative humidity levels and temperatures. There is no direct connection of this work and the costs of cardovascular pharmacotherapy, therefore the title is not justified by the content of the manuscript, and the first paragraph of the introduction is rather irrelevant (unless the authors provide some data or some model based estimate of the impact of ramipril's degradation on the costs of pharmacotherapy).
Response 1. The first paragraph of the Introduction has been removed.
Point 2. The materials and methods does not contain a clear description of the methods, but rather a reference to the instrumentation.
Response 2. Dear Reviewer, in section ‘2.3. Procedures’ the detailed description of the methodology and all the ana-lytical procedures were included. To our mind the description is provided in a way that allows the reproduction of the experiments. Furthermore, new sections 2.3.5. Reaction model-fitting and 2.3.6. Thermodynamic data calculation were introduced. Also in the section 2.3.4.1 and 2.3.4.2. the times of exposure were added.
Point 3. A description of how the relative humidity was regulated is missing.
Response 3. In the section 2.3.4. Kinetic study – preparation of samples the sentence “The experimental conditions were adjusted automatically” was added. Also in the section 2.2.3 ‘Other instruments’ the explanation was added as follows: ‘thermal chambers WAMED KBC – 125 W with automatic temperature and relative humidity (RH) control.
Point 4. Also, a description of the model fitting procedure should be added. How many and which kinetic models did the authors fit and why did they choose the first order model as the best fitting one?
Response 4. A new section 2.3.5. Reaction model-fitting was added. Also in the section 5.2 the following sentence was added: The reaction order was established mathematically by model-fitting method. The highest correlation coeffi-cient was obtained for the first-order reaction.
Point 5. A description of the transition state theory equations should be added in the methods, not in the results.
Response 5. A new section 2.3.6. Calculation of thermodynamic parameters was added.
Point 6. Confidence intervals are necessary in figure 4, as the lines overlap significntly.
Response 6. Dear Reviewer. Thank you for this very valuable remark. We agree that the Figure 4 was vague. We decided to include only one curve for the selected model mixture and the other ones were removed. We believe that now the Figure 4 clearly demonstrates how the tested excipients stabilize the API. In the Section 3.2.1. the fol-lowing sentence was added: ‘The correlation lnk = f(RH%) was presented graphically only for one model mixture since in the remaining cases the curves were analogous.”
Point 7. Are Figures 5, 6 and 7 necessary? They can be moved to the supplementary information.
Response 7. Dear Reviewer, as you recommend we decided to move Figure 5 and 7 to supplementary material
Point 8. As already mentioned, a better description of the transition state theory is needed. It is not clear how the ac-tivation energy is derived, by fitting the Arrhenious equation or by approximations of the parameters of tran-sition state theory equations.
Response 8. A new section 2.3.6. Calculation of thermodynamic parameters was added.
Point 9. What exactly is the novelty of this work? It is not unexpected that heat and moisture promote degradation of an API, neither it is unexpected that in presence of excipients, APIs degrade faster.
Response 9. Dear Rviewer, as we mentioned in the Introduction section, the stability data of drugs and APis remain confidential if the analyses are performed for regulatory purposes by manufacturers. The expiry dates of commercial drugs are claimed as short as possible. Thus, the detailed data about the kinetics of RAM degradation were unknown until now. The information we provided, that is the exact kinetic mechanism of degradation, enables the prediction of RAM sta-bility under any conditions of RH and temperature. Thus, the reliable scientific data for the extension of expiry date or confirmation of drug quality whenever it is necessary were given. The extended Arrhenius equation can be utilized with this aim. This in turn, provides the opportunity to store drugs for longer periods of time, for example to maintain federal strategic reserves. In this way, minimization of drug losses is achieved, which would be an effective strategy of cost reduction.
Furthermore the effect of excipients is, in fact, variable. There are excipients that stabilize API while others accelerate its degradation. Thus, the data provided in this manuscript indicate which excipients should be preferable for ramipril formulation.
The degradation impurities were identified for the first time. It was confirmed that the main degradation impurity in the presence of moisture is ramiprilat which deteriorates drug bioavailability (ramiprilat is not absorbed after oral admin-istration), yet it does not pose any safety concerns. The other impurity – the diketopiperazine derivative is formed only to a minor extent. Diketopiperazine derivative is formed in case of other structurally-related drugs and there is no toxicological data available for it. Here, the safety of ramipril was confirmed. This issue is also a novelty, as was shown by worldwide N-nitrosamine crisis in 2018 which confirmed that the industry does not analyse the degradation impurities thoroughly enough. We confirmed that no toxic degradation impurity is formed during ramipril storage.
The information on the storage mode of tablets is also extremely important. For pharmacists it comprises a basis for the promotion of good drug storage habits among patients. In fact, low patient compliance with label-storage recom-mendations is prevalent, which was confirmed in various research articles. A common, in-home practice involves use of weekly- or monthly-medication organizers, and storage of whole or halved tablets in damaged immediate packaging or even without immediate packaging under the high-moisture conditions. The above findings emphasize the im-portance of proper drug storage. It was shown for the first time that only commercial immediate packaging ensures the satisfactory protection from moisture, which seems to be the main reason for ramipril degradation. The loss of API during improper storage is impossible to detect visually because of absence of any physical changes in tablets’ ap-pearance.However, no studies were conducted to determine the interaction between the drug and excipients tested and the solid-state nature of the drug. I would recommend authors to address this comment and following minor comments before accepting the manuscript.

Round 2
Reviewer 1 Report
In my opinion, still the main problem is that I don’t see how this paper fits in the Pharmaceuticas journal when it is really analytical.
Author Response
Dear Reviewer,
Thank you for your comments. In the revised version of our manuscript we corrected English throughout the text in consultation with the native speaker.
Kind regards,
Katarzyna Regulska
Reviewer 3 Report
I would recommend accepting this manuscript in its present form.
Author Response

(The authors gave the same response as above.)

Reviewer 4 Report
The authors have responded to all points in a satisfactory manner. The manuscript is now suitable for publication.
Author Response

(The authors gave the same response as above.)
